# From Prebiotics to Probiotics: The Evolution and Functions of tRNA Modifications

**DOI:** 10.3390/life6010013

**Published:** 2016-03-14

**Authors:** Katherine M. McKenney, Juan D. Alfonzo

**Affiliations:** 1The Center for RNA Biology, The Ohio State University, Columbus, OH 43210, USA; anderson.2305@buckeyemail.osu.edu; 2The Ohio State Biochemistry Program, The Ohio State University, Columbus, OH 43210, USA; 3Department of Microbiology, The Ohio State University, Columbus, OH 43210, USA

**Keywords:** tRNA modification, evolution, genetic code

## Abstract

All nucleic acids in cells are subject to post-transcriptional chemical modifications. These are catalyzed by a myriad of enzymes with exquisite specificity and that utilize an often-exotic array of chemical substrates. In no molecule are modifications more prevalent than in transfer RNAs. In the present document, we will attempt to take a chemical rollercoaster ride from prebiotic times to the present, with nucleoside modifications as key players and tRNA as the centerpiece that drove the evolution of biological systems to where we are today. These ideas will be put forth while touching on several examples of tRNA modification enzymes and their *modus operandi* in cells. In passing, we submit that the choice of tRNA is not a whimsical one but rather highlights its critical function as an essential invention for the evolution of protein enzymes.

## 1. Before There Was RNA, There Were Modifications

Through the years there has been a seemingly limitless and constant supply of ideas and discussions about the origin of life; all envision numerous scenarios with the common theme that, based on modern day biological systems, replication and information storage were needed for a self-sustaining system to survive. In modern days, these functions are the realm of the nucleic acid/protein world. However, because of its rather “inert” (non-reactive) nature, DNA is thought to have appeared more recently and, in early life evolution, played a secondary role to that of RNA. The now well-established capacity for catalysis thus puts RNA at the front and center of the argument about the origin of life, a realization that dates back to the 1960s and was most recently verbalized under the term “RNA world” [1,2,3,4]. This was a time when the main transaction maker in biology was RNA, having the ability to replicate itself with high fidelity and little sequence specificity. Self-replication thus ensured the storage of critical genetic information while enabling the accumulation of variants with enough sequence diversity for evolution to run its course. Some variants catalyzed a diverse set of reactions, which, under the pressures of natural selection, led to a probably slow but sure transition to the mostly protein-catalyst world that we have today. Although the idea of an RNA world remains somewhat controversial with several complications that need explaining (*i.e.*, issues of chirality, selectivity for 5′-3′ linkages, the chemical instability of RNA, *etc.*), still the RNA world hypothesis offers the most plausible scenarios for what once was but is no more. In the present review, we will not attempt to make extensive arguments for or against various routes to an RNA world and will not even go into lengthy discussions of its pros and cons, rather we will assume the position that indeed such a world existed and was a required step in the origin of life. In that vein, we will present ideas and modern themes about the evolution of modified nucleosides, what functions they may have played in the RNA world, and in facilitating the appearance of tRNAs, their preferred targets. Ample evidence supports the fact that modifications were prevalent in pre-biotic times, thus we will also summarize the critical roles that such modified nucleotides played then and still play now and highlight their roles in affecting tRNA function in diverse biological systems.

## 2. Nucleotide Modifications in the Prebiotic World

Before there could be RNA, nucleotides had to exist. This is an inescapable dictum leading to the advent of the RNA world; tRNA included. Any discussions of the role modifications played at the beginning of life and what roles they play today, including their evolution and prevalence in tRNA, has to forcibly touch on what was there before life began. Such discussions inevitably start with what was possible in terms of prebiotic chemistry and the formation of nucleosides. One could imagine, and there should be little doubt, that certain basic elements/compounds were present in the early atmosphere including: CO_2_, H_2_O, N_2_, CH_4_, H_2_, and NH_3_ [5,6,7]. These, in various combinations, had to undergo condensation reactions that then led to other precursors such as formaldehyde (H_2_CO) and hydrogen cyanide (HCN). One major barrier in the early earth may have been the presence of an oxidized atmosphere, which could have prevented many of the reactions for prebiotic building blocks to occur readily, if at all [5]. Regardless, a few assumptions on substrate availability in the prebiotic earth, have led to questions of what is possible in terms of generating usable building blocks leading to nucleic acids. Most studies have focused on: (1) the prebiotic synthesis of the pyrimidine and purine bases; (2) the synthesis of ribose; (3) connecting the two to make nucleosides and nucleotides; and, lastly, (4) the formation of long enough polymers to give rise to catalytically active RNA and finally tRNAs; key to the transition into the protein-RNA world. Indeed, the 1950s and the famous Miller-Urey experiment marked the official birth of the field of prebiotic chemistry [8,9]. This was followed by now classical experiments in the works of Oró, Orgel, and Miller, which provided the first glimmers of evidence of what was possible [10,11]. Most significant was the early demonstration by Oró and Kimball of the formation of adenine from HCN, albeit aided by “contaminating” traces of glycolaldehyde in the HCN solution [10,11]. Soon after, various routes to form purines and pyrimidines were reported; most were logical variations on the Oró theme [10,11,12,13]. Thus, despite various arguments as to whether enough material was available in the early atmosphere for such products to be created in significant amounts, at least this early work opened a realm of possibilities.

The Miller group elegantly demonstrated that the early synthesis of nucleobases using chemicals likely found in the early earth not only produced the standard bases adenine, guanine, uridine and cytidine; it also generated a number of modified purines including 1-methyladenine, dimethylated purines, *etc.* [13]. In addition, pyrimidine derivatives were also generated most notably 2-thiolated uridine and cytidine [14]. This has led to several ideas as to what roles such modified nucleotides may have played in early life evolution. Cursorily, several articles have highlighted the fact that these modifications, if perpetuated, could have served to enhance the chemical diversity of oligoribonucleotides in the RNA world [15,16]. In the following sections, we will discuss what roles, beyond chemical diversity to increase catalytic capabilities, these modifications may have played while highlighting the functions that some of the most ancient modifications still play today.

## 3. Barriers to the Incorporation and Maintenance of Modifications

Amino acids clearly provide a larger chemical diversity than the four canonical nucleotides ever could, explaining the almost complete takeover of modern biological systems by protein enzymes. Nonetheless, with the discovery of naturally occurring ribozymes, it became apparent that RNA was not completely devoid of catalytic activities but the immediate question raised was as to what the limits of RNA catalysis were. This led to the revival of the old technique of *in vitro* evolution originally introduced by Spiegelman and co-workers but used now applying modern technologies [17,18,19,20,21]. Through *in vitro* selection it became clear that, despite obvious limitations of RNA as a catalyst, many reactions potentially important for the appearance of the RNA world were possible. For example, the selection of a ribozyme that could synthesize a nucleotide, another that could ligate two RNA oligomers, even the selection of an all-RNA replicase [22,23,24]. Beyond this, with the recent “rebirth” of the RNA modification field, some discussion of the roles that modifications may play in enhancing the chemical repertoire of ribozymes have been made, yet no easy route for *in vitro* selection schemes that efficiently exploit the chemical diversity provided by modified nucleotides has been devised.

One of the issues is that many post-transcriptional modifications are replication silent. That is to say, extant polymerases, such as the reverse transcriptase used in typical *in vitro* selection schemes, will erase any modifications at each round of selection and replace them for their canonical nucleotide equivalents. For example, pseudouridine becomes uridine, 2′-*O*-methylcytosine becomes cytosine, *etc.* Despite these complications, the fact that modified nucleotides were likely present in prebiotic times still begs the question as to what roles, if any, these played in the evolution of the RNA world. The obvious answer is in potentially increasing chemical diversity and with it the catalytic power of RNAs. This could have happened in *cis* via direct incorporation of modified nucleotides into a given RNA catalyst, whereby the modified nucleotide would be part of the active site of the ribozyme. However, this type of incorporation had to be position-specific and requiring persistence after replication (Figure 1). This we deem unlikely given the “erasing” power of replicases as discussed above. Alternatively, early ribozymes could have bound the modified nucleotide in *trans* at their active site, while positioning the modified side chain in the proper geometry for catalysis. This situation would be reminiscent of group I introns where the GMP nucleophile acts in *trans* [25]. In this realm, modified nucleotides would behave like co-enzymes for the early ribozymes. Indeed, it has been argued that the existence of nucleotide containing co-enzymes (e.g., those containing FAD for oxidation-reduction reactions) may well be remnants of the RNA world [26]. Perhaps a more relevant precedent is provided by the discovery of riboswitches that bind pre-Q1, a modified nucleotide precursor of queuosine [27].

We, however, would like to suggest a different possibility. It could well be that modified nucleotides played an important role not in catalysis but rather by altering RNA structure, whereby some modifications favor certain templates while others did not. Two aspects of modifications would then come into play: (1) the effect of some base modifications in altering nucleotide structure, both in free nucleotides as well as in the context of polymers; (2) some modifications may prevent base pairing, posing barriers to replication. It has been known for many years that some modifications may increase the flexibility of the ribose (*i.e.*, relax sugar puckering); other modifications, because of steric hindrance between the modified base and 2′ hydroxyl of the sugar, tend to make the sugar puckering rigid, almost strictly locking the sugar into a *C*-3′-endo conformation. For example, there is ample evidence showing that base thiolation (*i.e.*, 2-thiouridine, s^2^U) makes the sugar more rigid, while thiolation at a different position (4-thiouridine, s^4^U) makes it more flexible [28,29,30]. Likewise, base methylation may have similar effects. These effects are even stronger in the context of an RNA polymer, where the combination of steric hindrance between the base and the sugar and the base-stacking potential imparted in some nucleotides due to modification can further favor particular structures. The questions are: (1) how prevalent were such modifications in the prebiotic world, and (2) what role rigidity *vs.* flexibility could have played in the early RNA world? As a preface in answering these questions, it is worth mentioning that one of the great challenges for an RNA world has rested on the path that led to regiospecific selection of 5′-3′ linked polynucleotides *vs.* 5′-2′ linkages. It is clear that the generation of RNA polymers by non-enzymatic means leads to a mixture of 5′-3′ and 5′-2′ linked nucleotides in the context of an RNA polymer. It has been argued that although the presence of 5′-2′ linkages provide a degree of structural alteration, still at least with some *in vitro* selected ribozymes, between 10%–25% 5′-2′ linkages are tolerated to different degrees, in a context-dependent manner [31,32]. That is to say that local structural effects could be partly compensated by the presence of neighboring 5′-3′ linkages. In addition, the fact that 5′-2′ linkages destabilize RNA duplexes may have also served an adaptive advantage for strand separation prior to replication [31,32]. The reality still remains; however, that the predicted level of mixed linkages in prebiotic time may well have easily surpassed that 25% threshold begging the question as to how the RNA world managed to solve this riddle. Here we suggest that the presence of modified nucleotides in the prebiotic milieu may have tilted the scales to tolerable levels of mix linkages amenable to ribozyme activity. Notably, the prebiotic modifications discussed before such as thiolated pyrimides and methylated purines in the context of an RNA polymer almost exclusively favored a *C*-3′-endo conformation. It is plausible as proposed here that beyond the roles of modified nucleotides in enhancing chemical catalytic diversity, other roles in favoring replicable 5′-3′ linked polymers may be just equally important. One thing is clear; several factors may have contributed to the regioselectivity seen today in nucleic acid replication systems, which eventually required the invention of proteins and the establishment of the early genetic code.

## 4. Influence of RNA Modifications on the Evolution of the Genetic Code

Before the emergence of the three domains of life and early cell development, a primordial translation system had to evolve in the RNA world [33]. Along with the rudimentary translation apparatus, the genetic code, dictating which triplet codons correspond to which amino acid, had to be established. Transfer RNAs (tRNAs) are also assumed to have evolved early as essential mediators of the genetic code, becoming a link between information stored in messenger RNA (mRNA) and that delivered to proteins during translation. Initially, the basic translation system may have been imprecise in terms of the anticodon-codon pairing, reading-frame maintenance, efficiency and accuracy seen in modern translation systems [33]. The inaccuracy of the early translation machinery may have restricted the production of proteins to short peptides which likely acted as primitive enzymes [33]. Inaccuracy may have also served as a source of sequence diversity, providing necessary variants ripe for selection of “better” protein catalysts. Given the arguments for modified nucleotides in the primordial prebiotic milieu, eventually, the appearance of modified nucleotides in the early tRNAs was a certainty. RNA modifications then helped maintain the proper reading frame, facilitated the formation of longer peptides and led to higher enzyme complexity, while enhancing the accuracy of the system [33,34]. Ultimately, the evolution of tRNA modifications resolved critical translational issues prior to the split of the three domains Bacteria, Archaea, and Eukarya, as evinced by their universal presence [33,35].

Over one hundred RNA modifications have been discovered to date; the majority found in tRNA [36]. Of these modifications, eighteen are common to tRNA in all three domains of life and therefore considered primordial modifications [37]; these are relatively simple in terms of chemical makeup. The majority are comprised of methylations of the nitrogenous base or ribose sugar [37,38]. Other primordial modifications include addition of a small chemical groups such as the case of thiolation, or acetylation reactions. While still others are formed by a reduction of the C5-C6 double bond of uridine (dihydrouridine, D) or isomerization of the nitrogenous base (pseudouridine, ψ) [37]. Arguably, the most complex of the primordial modifications is the addition of the amino acid threonine to form N^6^-threonylcarbamoyladenosine (t^6^A). The biosynthesis of t^6^A involves the combination of threonine, carbonate, and ATP to form threonylcarbamoyl-adenylate (TCA) and the subsequent transfer of the TC group to tRNA [39,40]. This requires a multi-enzymatic pathway in all domains of life. Of the enzymes that catalyze this reaction, the enzyme families TsaC/Sua5 and TsaD/Kae1/Qri7 are conserved in all organisms throughout life, suggesting their presence in the last universal common ancestor (see recent review [41] for more detailed biosynthesis pathways). Over time, tRNA modifications continued to evolve and the appearance of more complex chemical moieties allowed for further fine-tuning of tRNA structure and function. More recently-evolved modified nucleotides (*i.e.*, those appearing after the split of the three domains) can be shared between two domains while some may be unique to a single one [37].

In general, tRNA modifications may affect translation in the following ways: by modulating tRNA structure and stability, achieving optimal rate of translation and translational fidelity/accuracy (correct codon recognition), or maintaining the reading frame [37,42,43,44,45,46,47,48]. Modifications present in the anticodon loop play significant roles in translational accuracy, efficiency, and/or reading frame maintenance [34,42,44,48,49,50,51,52]. In particular, the anticodon “wobble” position 34 and position 37, 3′-adjacent to the anticodon, are ubiquitously modified with a great diversity of different modifications, while modifications present in the body of the tRNA are usually important for proper folding and stability [42,44,49]. It is the former that may have been crucial early in the evolution of translational systems to help establish mechanisms for efficient high-fidelity translation.

## 5. Establishing Translational Accuracy

The universal genetic code consists of 64 potential codons, 61 of which are sense codons that specify 20 canonical amino acids and three of which are termination codons (UAA, UAG, UGA) [53]. The interaction between the anticodon of tRNA and the triplet codon of mRNA is fundamental to decoding the information stored in the nucleic acid and converting it to peptide formation. In other words, to achieve translational fidelity, each 3-nucelotide anticodon must pair with a 3-nucleotide cognate codon. The first two bases of the codon and second and third of the anticodon (positions 35 and 36) form canonical Watson-Crick base-pairs while the third position of the codon and first position of the anticodon (position 34) or “wobble” position may undergo degenerate (non-Watson-Crick) pairing. It is the base-pairing flexibility provided by the first anticodon position and the third position of codons that have allowed the degree of decoding flexibility and genetic code redundancy that still exists today, explaining why there are 64 codons for 20+ amino acids.

Some modifications endow tRNAs with the ability to expand their coding capabilities via wobble base-pairing, usually by providing alternative hydrogen bonding or base-stacking interactions which introduce structural flexibility or rigidity as needed [54,55]. Notably, although decoding flexibility is important in translating the modern genetic code, this flexibility has to be somewhat limited to prevent indiscriminate pairing of tRNAs to multiple non-cognate codons, risking the possibility of an ambiguous genetic code. Therefore, while some modification (e.g., inosine) enhance base-pairing, others restrict wobble base-pairing [54]. In all, tRNA modifications in the anticodon stem loop often act to discriminate between cognate, near-cognate, and non-cognate codons.

Of the primordial modifications, t^6^A at position 37 (t^6^A_37_), 2-thiouridine at position 34 (s^2^U_34_), and N4-acetylcytidine (ac^4^C) are important for accurate codon recognition [56]. tRNAs containing t^6^A_37_ are responsible for decoding mostly ANN codons and play a crucial role in AUG start codon selection [39,57]. The hydrophobic group of t^6^A_37_ forms a large planar structure ideal for stacking interactions and acts to reinforce binding of weaker A·U base-pairs [57]. The s^2^U modification has also been suggested to stabilize A·U and U·U base-pairs in RNA duplexes; an entropically driven process involving preorganization of a single strand before hybridization [58,59]. In tRNA, the steric influence of the thiol group forces the ribose sugar into the more thermodynamically favored C3′-endo conformation [58,60,61]. Studies in mutant strains of *Saccharomyces cerevisiae* showed that the presence in tRNA of s^2^U_34_ along with 5-methoxycarbonylmethyl-2-thiouridine (mcm^5^U_34_) enhanced pairing with A- and G-ending codons [62]. Moreover, *in vitro* binding experiments with tRNA^Lys^_UUU_ (subscript designating anticodon 5′-UUU-3′) demonstrated that t^6^A_37_, s^2^U_34_, as well as 5-methylaminomethyluridine (mnm^5^U_34_) are necessary for cognate pairing of codons ending in with A and G (purines) [56]. These modifications had originally been proposed to also restrict pairing with near-cognate codons ending with U and C (pyrimidines) [56]. However, this is clearly not always the case as recent evidence suggests that mnm^5^U_34_ and s^2^U_34_ do not appear to restrict base-pairing in yeast [63,64]. This is particularly true for tRNAs with pyrimidine-rich anticodons where pairing with another pyrimidine would be energetically unfavorable [64]. In bacteria, similar to what is seen with s^2^U, ac^4^C_34_ preferentially forms the C3′-endo ribose sugar pucker enabling stable pairing of tRNA^Met^_CAU_ to the AUG codon, while preventing pairing with the AUA codon [65,66].

After the divergence of the three domains, several different modifications appeared at the wobble position in addition to the primordial modifications discussed above. This evolutionary convergence emphasizes the functional importance of modifications at this position [67]. For example, post-transcriptional hydrolytic deamination of adenosine produces inosine, an analogue of guanosine, which is found in all domains of life at positions 34 (Eukarya and Bacteria), 37 (Eukarya), and 57 (Archaea) of tRNA. At positions 37 and 57, inosine undergoes further methylation to form m^1^I_37_, m^1^I_57_ or m^1^Im_57_, but their function has been largely unexplored [68]. In eukaryotes, 7 to 8 tRNAs contain I_34_, while in bacteria most of the corresponding tRNAs have G at position 34 and solely tRNA^Arg^ has I_34_. Replacing adenosine with inosine expands binding capabilities at the wobble position to A, C, and U, imparting profound influence over anticodon-codon recognition. Most importantly, these tRNAs require I_34_ to decode C-ending codons and it is therefore an essential modification in eukaryotes and bacteria.

While inosine increases decoding flexibility, the occurrence of potential ambiguities in the code had also to be addressed by modifications; for instance, the majority of sense codons ending with a purine correspond to a single amino acid with the exception of the AUA codon for Ile and AUG codons specifying Met. Decoding AUA and AUG codons as separate amino acids poses a challenge to organisms, especially since tRNA^Ile^_UAU_, even with a modified U at the wobble position, has the potential to read both NNA and NNG codons [69]. This decoding problem exists in all three domains of life, but different organisms have evolved independent modification strategies to solve it. Eukaryotes typically encode two isoacceptors of tRNA^Ile^, which have either inosine or pseudouridine at position 34. It is thought that both are able to decode AUA codons, but tRNA^Ile^_ψAψ_ is needed because decoding of A-ending codons by tRNA^Ile^_IAU_ is less efficient [70,71]. Whether tRNA^Ile^_ψAψ_ additionally prevents base-pairing with AUG codons is still unclear [69]. Bacteria and archaea also have two tRNA^Ile^ isoacceptors, tRNA^Ile^_GAU_ which binds C and G-ending codons and a tRNA with the modification lysidine (bacteria) or agmatidine (archaea) at position 34 [55,69,70]. Lysidine (2-lysylcytidine) and agmatidine (2-agmatinylcyditine) are modified cytidines with the ε-amino group of lysine or decarboxylated arginine conjugated at the C2 position of the base. The addition of the amino acid side chains alters hydrogen bonding capabilities of the nucleotide base, leading to an increase in base-pairing with adenosine while inhibiting base-pairing with guanosine [66,69]. Overall, these changes facilitate accurate decoding of AUA codons. With rare exception, the *tilS* and *tiaS* genes encoding the modification enzymes for lysidine and agmatidine, respectively, are essential [69]. Interestingly, a recent studies found that unmodified suppressor tRNA^Ile^_UAU_ from a *Bacillus subtilis tilS* deletion strain are still able to bind strongly to AUA codons and weakly to AUG codons [55,70]. This scenario is comparable to organisms devoid of both *tilS* and *tiaS* genes that are likely to have unmodified uridine at position 34 [69,72]. There are also organisms with both U_34_ and C_34_-containing tRNA^Ile^ as well as the *tilS* gene, supporting the evolution of U_34_ to a cytidine-derived lysidine modification [55].

Recently, the decoding properties of AUA and AUG codons have been exploited to introduce non-canonical amino acids into *E. coli* cells using an orthogonal translation system. Orthogonal translation systems typically rely on novel aminoacyl synthetase (aaRS):tRNA pairs in which the tRNA is assigned to a “blank” codon (usually a nonsense or frameshift suppressor) and the aaRS recognizes the tRNA and the non-canonical amino acid [73,74,75,76,77]. Several laboratories have taken advantage of the degeneracy of the genetic code by reassigning orthogonal pairs to decode sense codons, including the reassignment of rare AUA codons in *E. coli* [78,79,80,81,82,83,84,85,86]. For this experiment, an orthogonal aaRS:tRNA pair from *Mycoplasma mobile* a bacterium, which naturally lacks the tilS gene, was utilized. The *M. mobile* IleRS is able to recognize and efficiently charge unmodified tRNA^Ile^_UAU_ which is not the case in *E. coli* [72]. They were able to use this *Mm*IleRS:tRNA^Ile^_UAU_ pair to rescue lethal *tilS* deletion mutants in *E. coli* by direct AUA codon reading. The manipulation of tRNA modification pathways for the reassignment or “emancipation” of sense codons could potentially become an invaluable tool in the field of synthetic biology as this technique holds promise for genetic code expansion at a genome-wide level [78,83].

In addition to codon-anticodon pairing, fidelity of the genetic code can be influenced by aminoacylation of tRNAs. It is clear that, in order to maintain fidelity, the tRNA identity must reflect the genetic code. In addition to other structural and sequence identity elements, modifications such as t^6^A_37_, s^2^U_34_ (*E. coli*, not yeast), lysidine, and agmatidine are determining factors for correct charging by aminoacyl tRNA transferases (aaRSs) [62,87,88,89]. For example, unmodified tRNA^Ile^_CAU_ can act as a substrate for methionyl-tRNA transferase (MetRS) *in vitro* [70]. Once it has been modified with lysidine or agmatidine, it is no longer recognized by MetRS and instead is a substrate for IleRS [70]. Thus, lysidine and agmatidine in tRNA^Ile^ act as a determinant for both the charging of the right amino acid as well as enabling pairing with the correct AUA codon [69]. Conversely, other evidence suggests that unmodified *in vitro* synthesized tRNAs can act as adequate substrates for aaRSs implying that modifications are not always necessary for accurate charging [90].

## 6. Reading Frame Maintenance

While missense mutations caused by tRNA mischarging or inaccurate decoding can be detrimental for protein activity or structure, frameshift mutations are far more destructive to overall protein synthesis and are often fatal [35,51,52,91,92,93,94,95,96,97,98]. Studies of reading frame maintenance, as well as translational accuracy and efficiency, usually focus on A-site or P-site interactions within the ribosome. The A-site is where codon-anticodon recognition takes place whereas the P-site is where peptide transfer occurs. Frameshift errors can occur in the A-site, P-site, or both. This can happen if hypomodified, aminoacylated tRNA (in ternary complex with eEF1A in eukaryotes or EF-Tu in bacteria and GTP) enters the A-site and is mistaken for a cognate tRNA. After translocation to the P-site, the near-cognate peptidyl-tRNA does not result in an optimal fit causing it to slip [99]. Additionally, the inefficient acceptance of the near-cognate hypomodified tRNA into the A-site can induce ribosomal pausing which in turn may lead to peptidyl-tRNA slippage [99]. All in all, most cases of frameshifting initiated by hypomodified tRNA result from a reduction in the rate of A-site selection and/or inadequate accommodation in the P-site [99]. The ability of modifications to augment and fine-tune interactions within the A-site and P-site of the ribosome was imperative to the evolution of translation, exemplified by the function of primordial modifications like m^1^G_37,_ t^6^A_37_, s^2^U_34_ and pseudouridine in reading frame maintenance.

The modification m^1^G_37_ is critical for translational fidelity as it promotes selectivity of tRNA in the ribosomal A-site and prevents frameshifting [35,100,101,102,103]. m^1^G_37_ is found in nearly all tRNAs that pair with codons starting with C, except CAN codons [100]. Orthologous genes encoding m^1^G_37_ methyltransferase were identified from all three domains, implying that their presence dates back to the last common universal ancestor [35]. Furthermore, only three of greater than 500 sequenced tRNAs contained an unmodified G_37_; the prevalence of m^1^G_37_ illustrates its importance throughout life [35]. One of the first examples of the function of m^1^G_37_ in reading frame maintenance was shown in *Salmonella typhimurium*. This study utilized temperature sensitive mutants where m^1^G_37_ was completely absent in temperatures at or above 37 °C, resulting in increased frameshifting [100]. It was later suggested that the lack of m^1^G_37_ leads to decoding of a quadruplet rather than triplet codon causing +1 frameshifting [103,104]. More recently, however, conflicting structural evidence, along with the discovery that certain +1 frameshift suppressor tRNAs do not rely on Watson-crick base-pairing complementarity to function, has challenged the requirement of quadruplet base-pairing for frameshifting [104]. Based on structural studies, other potential modes of frameshifting induced by m^1^G_37_ deficiency have been proposed, including faulty translocation of the tRNA from A-site to P-site and P-site slippage. A critical U_32_-A_38_ base-pair, which is abolished when m^1^G_37_ is missing, has also been suggested to play a role in frameshifting [104]. Overall, without m^1^G_37_, the cell experiences global defects in translational fidelity but not in the efficiency of protein synthesis [100,103,104]. Moreover, down-regulation of the m^1^G_37_ methyltransferase Trm5 severely affects mitochondrial protein synthesis and function in trypanosomes [101]. This observation supports earlier reports of respiratory defects upon depletion of m^1^G_37_ in mitochondrial initiator tRNA^Met^ in yeast [105].

In addition to their roles in accurate decoding and charging by aaRSs, t^6^A_37_ and s^2^U_34_ are also important in maintaining the proper reading frame. In *S. cerevisiae*, loss of one of the t^6^A_37_ enzymes, Sua5 or Kae1, leads to increased frameshifting [57,106]. Along these same lines, a recent analysis of wobble uridine modifications using a combination of reporter systems in *S. cerevisiae* deletion mutants lacking U_34_ modifications revealed that s^2^U_34_ and mcm^5^U_34_ prevent +1 frameshifting in tRNA^Lys^ and tRNA^Glu^ [99]. Additionally, a study using kinetic assays showed that thiolation influenced decoding and accommodation as measured by GTP hydrolysis and peptide bond formation, respectively [107]. The incorporation of s^2^U_34_ into *E. coli* tRNA^Gln^_UUG_ increased the decoding rate of both cognate and near-cognate codons by five-fold [107]. Taken together, it has been proposed that the +1 frameshifting resulting from the absence of s^2^U_34_ may be at least partially due to the impact of thiolation on translational efficiency, particularly impacting efficient acceptance into the A-site [99].

The universal and highly abundant tRNA modification, pseudouridine, has also been implicated in reading frame maintenance. In *S. cerevisiae*, the deletion of the *pus3* gene, encoding a pseudouridine synthase responsible for modifying positions 38 and 39, resulted in pronounced translational and cell growth defects [108]. By measuring the efficiency of translational read-through and frameshifting via a dual reporter system, ψ_38_ and ψ_39_ in certain tRNAs were found to promote proper codon reading and an increase in +1 frameshifting [108]. A similar effect on reading frame maintenance, although in the −1 direction, was observed in a related study using a yeast *pus3* deletion strain [109]. In either case, it is clear that ψ_38_ and ψ_39_ impacts the translational reading frame.

## 7. Evolution of tRNA Modification Enzymes

The enzymes responsible for a subset of modifications important for translation are either conserved or have undergone functional convergence. Conserved tRNA modification enzymes have evolved from a common ancestor and have similar sequence and structure. Other tRNA modifications are catalyzed by enzymes that are evolutionarily unrelated but serve the same function. An example of the former are the conserved dihydrouridine synthases (Dus), a family of enzymes which catalyze the formation of the primordial modification, dihydrouridine. Dihydrouridine is one of the most abundant modified nucleosides in tRNA, usually found in the D-loop (for which it is named) and at one position in the variable loop [110,111]. Dihydrouridine synthases utilize flavin mononucleotide (FMN) as a cofactor along with NADPH as the electron donor to reduce the C5-C6 double bond of uracil to generate dihydrouridine. The resulting dihydrouridine forms a nonplanar base which prevents favorable stacking interactions and promotes C2′-endo rather than the typical C3′-endo ribose conformation. These alterations destabilize local stacking interactions and are thought to introduce conformational flexibility into the tRNA [110,111,112,113]. Genes encoding Dus enzymes are ubiquitous, having been found in all organisms sequenced thus far [111]. Three Dus enzymes have been identified in *Escherichia coli* (DusA, DusB, and DusC) via a bioinformatics approach and four in yeast (Dus1p, Dus2p, Dus3p, and Dus4p) through biochemical assays [110,111,114,115,116]. Based on a comprehensive bioinformatics analysis, it appears that modern Dus enzymes evolved from a single ancient Dus that underwent independent duplication in Eukarya and Bacteria, but not Archaea [111].

Interestingly, the catalytic center of all characterized Dus enzymes have a conserved active site and the same overall fold, but most modify specific and spatially distinct positions on tRNA [112,115,117]. For example, in *S. cerevisiae* Dus1p modifies position 16 and 17, Dus2p position 20, Dus3p position 47, and Dus4p positions 20a and 20b [113,115]. Likewise, the *E. coli* Dus enzymes act on non-overlapping substrates [110,117]. The ability to act on a wide array of nucleotide positions within tRNA while maintaining a conserved global fold raises the question of how these enzymes recognize and precisely modify their substrates. The crystal structure of *Thermus thermophiles* Dus bound to tRNA^Phe^ revealed that, upon binding, the enzyme distorts the D-loop but maintains and requires critical D-loop/T-loop interactions. This suggests the enzyme monitors the overall canonical L-shape of the tRNA [113]. More recently, the crystal structure of the *E. coli* DusC enzyme specific for U_16_ alone or complexed with two different tRNAs has been reported [112,118]. It has been proposed that U_16_- and U_20_-specific Dus enzymes bind tRNA in different orientations directed by putative “binding signature” amino acid sequences (Figure 2). The substrate is then further adjusted by a the C-terminal “recognition” domain guiding the tRNA in the appropriate position in the active site [112]. Therefore, it appears that while the overall fold and activity of the Dus enzymes are conserved, each family has evolved a unique way of tRNA docking to ensure substrate specificity.

Since the modification, m^1^G_37_, is an ancient modification with significant consequences in translation (as highlighted earlier in this review), it is reasonable to expect that the enzymes that catalyze this reaction would be evolutionarily related. On the contrary, Trm5 methyltransferases from eukaryotes and archaea exhibit an entirely different fold and substrate binding from the bacterial TrmD methyltransferase indicating that they evolved independently [38,119]. As a result of considerable biochemical, bioinformatics, and structural research, five different structural folds (I–V) described that use *S*-adenosyl-l-methionine (SAM or AdoMet) as a methyl donor to act on a diverse array of substrates [119]. The majority of SAM methyltransferases fall under the Rossmann-fold class I category, which is the case for Trm5 enzymes (Figure 3) [120]. Class I methyltransferases are comparable to canonical Rossmann-folds consisting of a parallel β-sheet surrounded by helices, except they typically have an additional (seventh) antiparallel β-strand [121]. While oftentimes Rossmann-fold methyltransferases vary widely in sequence, they generally contain a GxGxG, characteristic of a nucleotide binding site, which bends underneath SAM to form the first α-helix [122]. Similar to the Dus enyzmes, structural studies suggest that Trm5 identifies the canonical l-shape of the tRNA in its substrate selection process [119,123]. Unlike Trm5, the bacterial m^1^G_37_ methyltransferase, TrmD, is a class IV fold or SPOUT (SpoU or TrmH and TrmD) family methyltransferase defined by its classical trefoil knot SAM-binding motif (Figure 3). The deep trefoil knot is shaped by folding the C-terminus of the SAM-binding domain into the catalytic pocket [119]. According to the crystal structure of the TrmD homodimer complexed with tRNA and a SAM-analogue, along with biochemical analysis, TrmD recognizes solely the tRNA anticodon branch including interactions with the D-loop [119,124]. However, it does not make contact with the acceptor stem or T-loop [119,124]. Thus, Trm5 and TrmD are composed of disparate N-terminal SAM-binding domains and C-terminal tRNA recognition domains. (For a recent review of tRNA methyltransferases, see Reference [38]).

## 8. Modularity of tRNA Modification Enzymes

In general, most enzymes are modular in nature and are known to acquire domains that become critical for substrate recognition, binding or catalysis. The tRNA modifying enzymes are not an exception; some enzymes recognize the global structure of the tRNA and exhibit a broad range of substrate binding (as previously mentioned with Dus and Trm5 enzymes), others have recruited RNA binding domains [125]. These RNA binding domains are commonly fused with catalytic domains and confer specificity to a reduced set of tRNA substrates. One such example of the enlistment of RNA binding domains is seen in tRNA editing enzymes that catalyze the ubiquitous A-to-I deamination. Adenosine deamination is the most common form of RNA editing and it is not restricted to tRNA. Two major families of editing deaminases exist, adenosine deaminases acting on RNAs (ADARs) and polynucleotide cytidine deaminases (CDAs), which are categorized based on their active site and zinc metal-binding sites. The eukaryotic m^1^I_37_ deaminase enzyme, ADAT1 or Tad1 resembles classical ADARs except without a double-stranded RNA binding domain (dsRBD) (Figure 4) [126,127]. This close similarity may give insight into the evolution of ADARs whereby an ADAT1-like enzyme obtained a dsRBD, enabling the switch to mRNA editing [127]. In fact, protein sequences of ADAT1 from higher eukaryotes are more similar to ADARs than to yeast Tad1, further supporting this evolutionary hypothesis [127].

Surprisingly, the enzymes responsible for I_34_ formation in Bacteria (TadA) and Eukarya (ADAT2/3 or Tad2/3) are more closely related to CDA family enzymes rather than ADARs [128,129]. The CDA family contains the sequences H(C)XE and PCXXC (X represents any amino acid) which form the active site of the enzyme with a catalytic glutamate and zinc coordinating residues (Figure 4) [130]. In eukaryotes, ADAT2 harbors the critical active site glutamate (HAE) while ADAT3 has substituted for a non-catalytic residue (HPV in *T. brucei* and *S.* cerevisiae) and was originally thought to play a purely structural role [131]. However, inductively coupled plasma emission spectrometric (ICP) analysis of the enzyme from *T. brucei* demonstrated that both subunits contribute to zinc binding, suggesting both ADAT2 and ADAT3 are essential for catalysis [132]. Along with the active site, we found that ADAT2 in *T. brucei* has acquired a lysine and arginine-rich tRNA binding domain, named the KR-domain, at its C-terminal end [130]. Presumably, the density of positive charges interacts favorably with the phosphate backbone of the tRNA. A highly charged domain corresponding to that seen in ADAT2 has also been described in other deaminase enzymes, including ADAR3 [130]. The discovery of the KR-domain supports a previous model whereby the C-terminus of ADAT2 in Eukaryotes evolved a conserved tRNA binding domain expanding its substrate repertoire to include eight different tRNAs [133].

In addition to substrate binding, the presence of critical modular catalytic domains is also common, such as SAM-binding domains of methyltransferases and FMN-binding domains of Dus and enzymes that form wybutosine and hydroxywybutosine. Wyosine and its derivatives, including wybutosine (yW) and hydroxywybutosine (OHyW), are critical for translational fidelity in archaea and eukaryotes, but have not yet been described in bacteria. These highly complex modifications are found exclusively at position 37 of tRNA^Phe^ where the bulky hydrophobic side chain stabilizes the anticodon loop structure, as well as anticodon-codon pairing, preventing potential -1 frameshifting [134,135]. The biosynthesis pathways for these modifications differ greatly between the two domains. While all wyosine derivatives require m^1^G_37_ as a precursor, eukaryotes employ a series of enzymes designated TYW1 through TYW5 whereas the archaeal pathway is comprised of enzymes TAW1 through TAW3. However, in both Eukarya and Archaea, various combinations of these enzymes leads to the formation of different derivatives of wyosine depending on the organism. For instance, most eukaryotes carry out sequential reactions by TYW1 through TYW4 with wybutosine as the end product [123,135,136,137,138]. Meanwhile, in other eukaryotic organisms, such as humans, wybutosine can be further modified to hydroxywybutosine by TYW5. The situation in archaea becomes more complex as the enzymes TAW1, TAW2, TAW3 carry out their chemistries either sequentially or in different combinations to generate wyosine derivative ranging from 4-demethylwyosine (imG-14) to 7-aminocarboxypropylwyosine (yW-72). For instance, Crenoarchaeota lacking TAW2 combine TAW1 and TAW3 to yield wyosine as the final product. The key reaction to form the unique tricyclic core characteristic of wyosine nucleosides is catalyzed by the eukaryotic TYW1 and its archaeal counterpart TAW1. The resulting 4-demethylwyosine is an intermediate in the eukaryotic multienzymatic biosynthesis pathway and can act as a final product in archaea [139]. Once guanosine has been converted to m^1^G by Trm5, TYW1/TAW1 adds two carbons derived from pyruvate to create an imidazole ring [139,140]. An N-terminal flavodoxin-like domain and C-terminal radical SAM catalytic domain, comprised of a characteristic CxxxCxxC iron-sulfur (4Fe-4S) binding motif, enables the chemistry performed by TYW1 (Figure 4) [141,142]. Although it has not been investigated, the FMN prosthetic group undoubtedly participates in reduction of Fe-S clusters, a function which is required in radical-mediated SAM reactions [142]. Unlike TYW1, TAW1, has no FMN-binding domain and must obtain reducing power from an outside source (Figure 4) [143].

A phylogenetic analysis of archaeal TAW1 and eukaryotic TYW1 gene sequences revealed that a gene duplication event likely occurred whereby the eukaryotic lineage acquired an FMN-binding domain [144]. The possibility of an ancestral bacterial enzyme is discounted due to the fact that bacteria do not have wyosine. Likewise, horizontal transfer of the eukaryotic enzyme from Archaea is unlikely as they did not cluster together in such an analysis [144]. Remarkably, *T. brucei*, has two paralogous TYW1 enzymes, one localized to the cytoplasm and the other to the mitochondrion. *T. brucei* often displays unusual biochemistry distinct from most eukaryotes as it belongs to an early-branching group of unicellular organisms, called kinetoplastids. It is the only organism described thus far with wyosine in the mitochondrion. The cytosolic enzyme, TYW1L, has the FMN binding domain, two 4Fe-4S cluster domains, a radical SAM domain, and a “wyosine base formation motif” (Figure 4). The mitochondrial enzyme, TYW1S, is reminiscent of the archaeal TAW1 and contains all the previously mentioned domains except for the FMN binding domain [144]. All in all, it appears as though the cytosolic TYW1L in *T. brucei,* akin to other eukaryotes, has more recently acquired the FMN-binding module. The particular example of the wyosine system in Eukarya and Archaea yet provides a unique testament to the general modularity of tRNA modification enzymes and how such modularity may have evolved.

## 9. Concluding Remarks

In the present review, we have highlighted certain aspects of post-transcriptional modifications using tRNA as the center piece of our discussions. We hope it is clear that modifications have played key roles in the evolution of living systems. These roles arguably predated protein-catalyzed reactions that are signatures of extant biological systems. Through studies that reproduce what was possible in early chemistry, it was demonstrated many years ago that prebiotic chemistry, and non-enzymatic synthesis, inevitably led to the formation of modified nucleosides in quantities nearly equimolar to those of the four canonical ones. Such observations have provided strong arguments for the antiquity of modifications on earth leading to the conclusion that “modified nucleosides always were” so eloquently stated by Cedergren and co-workers [145]. Using prebiotic chemistry as the starting point, we have tried our best here to discuss features of modifications that may have contributed to the appearance of an RNA world; a forcible intermediate to the evolution of biological systems now dominated by protein enzymes. However, even today, a number of biological catalysts cannot shake their RNA habit and RNA function remains a necessary component of modern biology. From the tRNA point of view, we have also provided examples where modifications have played critical functions in establishing and molding the genetic code and in parallel discussed aspects of the evolution of modification enzymes themselves.

In concluding, we highlight newer facets of modifications that go beyond their immediate role in translational fidelity and optimization of protein synthesis efficiency. A growing theme in the field is now the roles and connections between metabolism and cell homeostasis mediated by tRNA post-transcriptional modifications. In this realm, there is perhaps no better example than the modified nucleotide queuosine, synthesized by bacteria and salvaged by most eukaryotic organisms [37,48]. Although its full significance to cell physiology is not fully understood, there is growing evidence that reduced levels of queuosine correlate with cellular malfunctions, for example in many types of cancers in mammals or short longevity and sterility in flies [146,147,148,149,150,151,152]. Thus, queuosine may well be one of the original probiotics; bacteria make it and we take it. Makes one wonder how many other such micronutrients bacterial commensals produce that play critical roles in cellular physiology but have escaped our attention. We thus hope that our review makes sufficiently strong arguments for a new appreciation of the fact that modified nucleotides “always were” but also of their importance today.

## Figures and Tables

**Figure 1 life-06-00013-f001:**
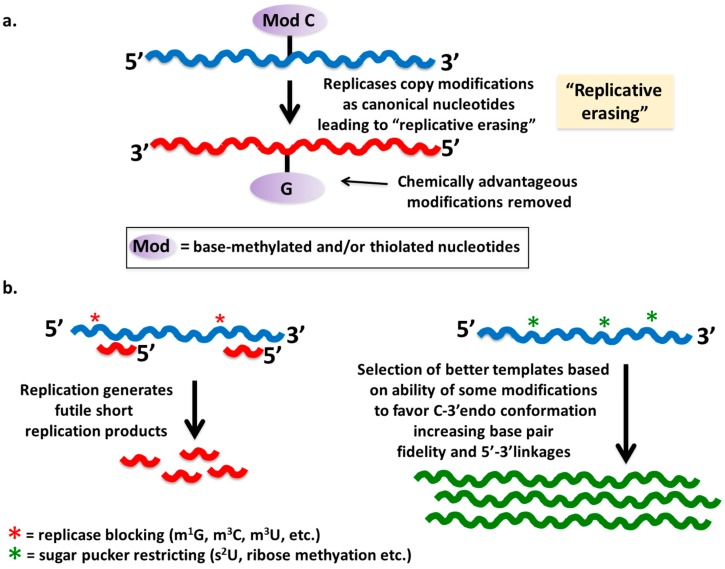
Potential outcomes and barriers for the incorporation and maintenance of modified nucleotides in the RNA world. (**a**) A main barrier for exploiting the increased chemical diversity of modified nucleotides in the RNA world would have involved the idea of “replicative erasing” whereby modifications are systematically removed (“erased”) from a given template. This is made possible by the mere fact that most replicases are blind to the presence of modified nucleotides and systematically replace them for their canonical nucleotide equivalent. (**b**) Upon incorporation, modifications may; however, provide a selective advantage either by preventing replication (scheme on the left) or by favoring a given sugar puckering (for example C-3′ endo *vs.* C-2′endo) which in turn may favor base pair fidelity and/or 5′-3′ linkages as proposed (scheme on the right). “mod” refers to any number of modified nucleotides available though pre-biotic synthesis. Asterisks denote the presence of certain types of modifications whereby red asterisks denote modifications that block base pairing thus blocking replication; green asterisks denote modifications that favored rigidity in sugar puckering.

**Figure 2 life-06-00013-f002:**
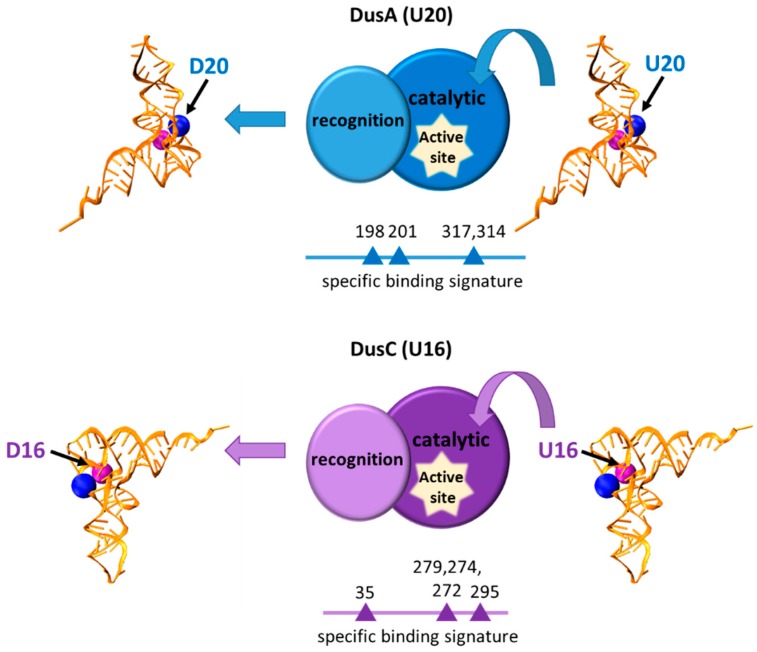
Model representing proposed binding specificity of *E. coli* dihydrouridine synthases, DusA and DusC. The location of the target uridine for each enzyme are shown as spheres on the tRNA in blue (U16) and purple (U20). The relative location and amino acid residue number of the signature binding positions are indicated below each enzyme. Binding signatures unique to each sub-family of Dus enzymes allow for major-reorientation of substrate tRNA while they exhibit a conserved overall structure, including a recognition and catalytic domain.

**Figure 3 life-06-00013-f003:**
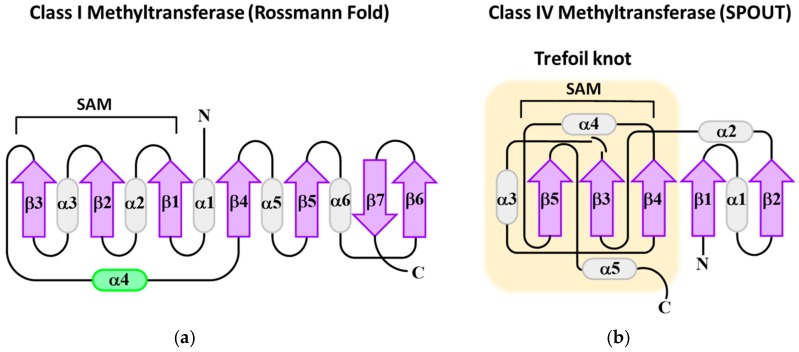
Topology diagram of Class I (**a**) and Class IV (**b**) methyltransferases. The alpha helices are shown as cylinders and beta sheets as arrows. The alpha helix in green is not always conserved in Rossmann fold methyltransferases. The SAM binding domains are indicated by brackets. The characteristic trefoil knot motif of SPOUT methyltransferases is highlighted in yellow.

**Figure 4 life-06-00013-f004:**
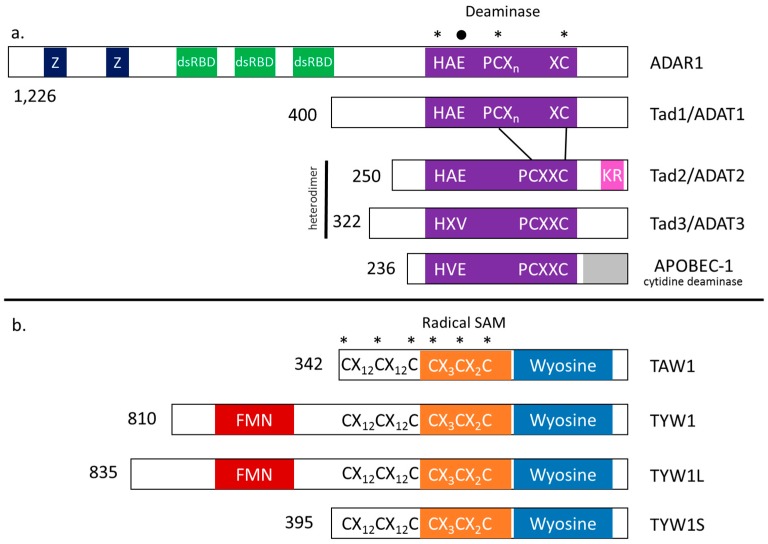
Modularity of deaminase and wybutosine enzymes. Domain organization of (**a**) Adenosine deaminase enzymes: ADAT1 (human), Tad1/ADAT1 (*S. cerevisiae*), Tad2/ADAT2 (*S. cerevisiae* and *T. brucei*), and Tad3/ADAT3 (*S. cerevisiae* and *T. brucei*) and cytidine deaminase enzyme: APOBEC-1 (human). The deaminase domain (purple) contains an active site comprised of characteristic Zn^2+^-binding residues (denoted by asterisks, *****) and a critical proton-shuttling glutamate (denoted by a filled circle, •). Z-DNA (dark blue) and double stranded RNA binding domains (green) are unique to ADAR1. Tad2/ADAT2 contain a KR-rich tRNA binding domain (pink) and APOBEC-1 has a pseudoactive site (grey) at the C-terminal end; (**b**) Enzymes which catalyze 4-demethylwyosine (imG-14), the second step of wyosine formation in Archaea: TAW1 (*Methanocaldococcus jannaschii*), and wybutosine formation in Eukarya: TYW1 (*S. cerevisiae* ), TYW1L (*T. brucei*), TYW1S (*T. brucei)*. The wyosine formation domain typical of this family of enzymes is shown in blue. Iron-sulfur cluster (4Fe-4S) coordinating residues (shown by asterisks, *) are found within and just before the radical SAM domain (orange). The flavin mononucleotide (FMN) domain (red) is found in TYW1, as well as TYW1L, but not TAW1 or TYW1S. The protein length in amino acids is indicated by the numbers at the N-terminus of the protein.

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
