# Peer review of "From Prebiotics to Probiotics: The Evolution and Functions of tRNA Modifications"

_life, 2016, doi:10.3390/life6010013_

Round 1

Reviewer 1 Report

The manuscript by McKenney and Alfonzo presents a balanced and comprehensive overview of the entire tRNA modifications and it is worth to be published in the current form (after removing some typos). The authors provided convincing examples of modifications that have played critical roles in defining the modern genetic code and understand a framework for the development of tRNA modification enzymes.

I would be happy to support full publication of this paper provided that the authors in their discussion (in the revised Ms) on the decoding issues of AUA and AUG codons also include recent report on experimental attempts to 'emancipate' AUA codons for genetic code expansion in Escherichia coli (FEMS Microbiol Lett ., 351, 133-144, 2014).

Namely, the issue of tRNA modifications will most probably become critical in future designs of orthogonal translation as the focus will doubtlessly shift on exploiting the degeneracy of the genetic code for the reassignment of rare sense codons.

Reviewer 2 Report

Authors discuss several features of tRNA modifications that may have contributed to the appearance of the RNA world and its evolution. They comment several works supporting that modified nucleosides were possible in prebiotic times. They also propose that tRNA modifications may have influenced the evolution of the genetic code by enhancing the accuracy of the translation and, consequently, facilitating the production of better protein catalysts. In this way, the evolution of tRNA modifications could resolve critical translational issues prior the split of the three domains of life, which is supported by the universal presence of eighteen modifications (named primordial modifications). In this point, the authors describe the role of several primordial modifications in translational accuracy and reading frame maintenance. In the last part of the article (points 7 and 8), the authors provide examples of the evolution and modularity of tRNA modification enzymes.

This review is hard to read, mainly the first part (points 3-6), because it addresses several functions of tRNA modifications without supporting material (i.e. figures and/or schematics), tries to cover too many aspects in a few pages, and mix information on primordial modifications and more complex modifications. However, the review may be useful for experts in the RNA modification field, as it provides: 1) interesting ideas on what functions modifications have played in the RNA world; 2) data on which roles they currently play in diverse biological systems; and 3) the evolution and modus operandi of several tRNA modification enzymes.  

Specific comments:

Authors state that primordial modifications are relatively simple in terms of chemical makeup (p. 4, lines 175-176), but t6A appears to be a complex modification. I think that the authors should provide some information on the synthesis on this nucleoside.

In relation to the role of s2U (p. 5, lines 219-220), reference 46 is incomplete. Moreover, the role of s2U is clarified in NAR 43: 7675, 2015.

In p. 5, lines 225-227, the authors state that mnm5U restricts pairing with near cognate codons ending in pyrimidines but this is not right at least for the case of tRNA-Lys (Nat Struct Mol Biol 11: 1186, 2004; see last paragraph p.1190, left column).

Other points:

Reference 38 is incorrect (RNA Biol 12: 398-411, 2015). Authors should review all references.

Authors should correct typos throughout the text.

Use s2U, not S2U, as abbreviation.

Reviewer 3 Report

In their excellent review McKenney and Alfonzo discuss (t)RNA modifications from an evolutionary perspective covering both the origin of modified nucleotides in RNA and the evolution of tRNA modifying enzymes (by examples).

The review represents a modern view on a very interesting topic that has not been covered recently by similar reviews. It is therefore of interest to the RNA modification field and evolutionary biology and translation research.

There are some minor inaccuracies:

Line 26/27 vs. line 36: Nucleic acid/protein word vs. RNA/protein world that we have today. This is a contradiction. Please smoothen.

Line 39: remove “it”

Line 105: remove “did”

Line 129: Ref. 27 seems wrong here.

Line 184-186: Efficiency is not the ideal term here. Certain protein/RNA sequences may require “low efficiency” which is in essence low speed to coordinate the ribosome with the folding machinery

Line 187+190: Is Ref. 39 indeed the correct one?

Line 214: t6A is considered a primordial modification. It is chemically also the most complex one. This is counterintuitive. Please speculate why is it a primordial modification.

Line 224-230: Ref. 49 and in vitro data is in contrast to recent work by Nedialkova D et al. 2014 where wobble U modifications don’t matter for G ending codons.

Line 231: “myriad of modifications”: Many of those are just xm5U variants. Do the authors really consider those cases as “different modifications”. Please explain and tone down.

Line 271/272: s2U is not a determining factor for (aaRSs) in yeast Johansson, et al. 2008.

Line 526 “synthesized”

Line 619: Ref 43. In odd format?

Line 770/771: year missing from reference. (please check all references from books)

Reviewer 4 Report

This very well written manuscript provides a novel perspective on modified ribonucleosides in tRNA in the context of evolution of life. The authors make a series of well grounded arguments about the emergence of modified ribonucleosides and the evolution of their different functions in tRNA and translation biology. Of course, one can mount counterarguments for some of the authors’ proposals. However, these kind of perspective pieces are necessarily speculative and the authors present interesting and provocative models and arguments for which there is some experimental support. This paper will be of interest to readers of this special issue of Life.
